# Independent prescribing by advanced physiotherapists for patients with low back pain in primary care: A feasibility trial with an embedded qualitative component

Tim Noblet[1,2‡]*, John Marriott[3‡], Amanda Hensman-Crook[4‡], Simon O'Shea[5‡], Sarah Friel[6‡], Alison Rushton[1‡]

1 Centre of Precision Rehabilitation for Spinal Pain, School of Sport, Exercise and Rehabilitation Sciences, University of Birmingham, Edgbaston, Birmingham, England, United Kingdom, 2 Physiotherapy Department, St George's University Hospitals Foundation Trust, London, England, United Kingdom, 3 Institute of Clinical Sciences, College of Medical and Dental Sciences, University of Birmingham, Edgbaston, Birmingham, England, United Kingdom, 4 Windermere Health Centre, Windermere, England, United Kingdom, 5 Sheffield Teaching Hospitals NHS Foundation Trust, Sheffield, England, United Kingdom, 6 Guys and St Thomas' NHS Foundation Trust, London, England, United Kingdom

‡ TN is the primary author. JM and AR contributed equally to this work as second authors. AH-C, SO, and SF contributed equally to this work as site primary investigators.
* timnoblet@hotmail.com

**Data Availability Statement:** The data underlying the results presented in the study are available from CPR Spine, School of sports, exercise and

## Abstract

### Background

Low back pain (LBP) is the most prevalent musculoskeletal condition. Guidelines advocate a multimodal approach, including prescription of medications. Advanced Physiotherapy Practitioners (APPs) are well placed to manage LBP. To date no trial has evaluated the efficacy of physiotherapist-prescribing for LBP.

### Objectives

To evaluate the feasibility, suitability and acceptability of assessing the effectiveness of physiotherapist-prescribing for LBP in primary care; informing the design of a future definitive stepped-wedged cluster trial (SWcRCT).

### Methods

Mixed-methods, single-arm feasibility design with two components.

1) Trial component: participants with medium-risk LBP +/-leg pain were recruited across 3 sites. Outcome measures (primary outcome measures-Pain/RMDQ) were completed at baseline, 6 and 12 weeks Physical activity/sedentary behaviour were assessed over 7 days using accelerometry. A CONSORT diagram analysed recruitment/follow-up rates. Descriptive analysis evaluated procedure/floor-effects.

2) Embedded qualitative component: focus groups (n = 6) and semi-structured interviews (n = 3) evaluated the views/experiences of patients and APPs about feasibility/suitability/

rehabilitation sciences, University of Birmingham. Please email: CPR.Spine@contacts.bham.ac.uk.

**Funding:** Health Education England (HEE) funding has allowed for the procurement of accelerometers and the associated IT programmes to ensure that innovative physical measures can be evaluated alongside patient reported outcome measures. The Private Physiotherapy Educational Fund has allowed for the procurement of x3 tablet computers for use in data collection and 7.5hrs per week of the principal Investigators time for 18 months. The funders have no direct role in study design, data collection and analysis decision to publish, or preparation of the manuscript. There were no conditions attached to funding. Identification of the trial funders provides transparency and accountability.

**Competing interests:** All authors have completed the ICMJE uniform disclosure form at www.icmje. org/coi_disclosure.pdf and declare that they have no competing interests: no support from any organisation for the submitted work; no financial relationships with any organisations that might have an interest in the submitted work in the previous three years; no other relationships or activities that could appear to have influenced the submitted work.

acceptability of the proposed trial. Thematic analysis synthesised the qualitative data. Findings were evaluated against a priori success criteria.

## Results

n = 29 participants were recruited. 90% of success criteria were met. Loss to follow-up at 12 weeks (65.5%) did not satisfy success criteria. Primary and secondary outcome measures were suitable and acceptable with no floor effects. The addition of a sleep assessment tool was advised. Accelerometer use was acceptable with 100% adherence. APPs felt all patients presenting with non-specific LBP +/- leg pain and capture data representative of the full scope of physiotherapist independent prescribing should be included. Data collection methods were acceptable to APPs and patients. APPs advocated necessity for using research assistants owing to time limitations.

## Conclusions

Methods evaluated are feasible, suitable and acceptable for a definitive SWcRCT, with modification of eligibility criteria, and use of research assistants to overcome limited clinician capacity. A definitive SWcRCT is feasible with minor modifications.

## Registration

ISRCTN15516596.

## Background

In the UK, over 30 million working days are lost per year owing to musculoskeletal conditions [1]. Low back pain (LBP) is the most common musculoskeletal disorder, with 28.5% of the population over 25 years old experiencing LBP at any one time [2]. Seven percent of the UK population experience LBP associated with high levels of disability [2, 3]. Despite advances in knowledge, understanding and awareness regarding the complex biopsychosocial nature of this prevalent and multifaceted problem, the health and function of those with LBP continues to decline [4]. Novel approaches to the assessment and management of LBP, such as the use of stratification tools and shared decision-making, have been introduced across health sectors in an attempt to reduce overall costs to the health economy. It is theorised that by ensuring that biopsychosocial risk factors are assessed and managed appropriately, patient outcomes will improve and the burden on the health system will reduce [4–6].

Early management of LBP is reported to reduce the likelihood of chronicity [3]. 20% of adults with LBP seek help in primary care, equating to 7% of all general practitioners' (GP) consultations [7, 8]. The population of the UK is growing and the mean age is increasing; contributing to the current deficit in GP availability [9, 10]. It is predicted that the number of GPs will further decline by 2020 [11, 12]. Following successful pilot studies testing innovative clinical pathways to optimise clinical and cost-effectiveness [10, 12, 13], NHS England have mandated the introduction of musculoskeletal first contact practitioners (FCPs) in primary care. This innovation aims to enable timely access to specialist musculoskeletal practitioners such as Advanced Practice Physiotherapists (APPs), without the patient first seeing a GP [11, 12].

APPs have been shown to be clinically and cost-effective and are experts in the assessment, diagnosis and management of musculoskeletal conditions [14–16]. APPs have traditionally

worked as specialist clinicians in primary/secondary care musculoskeletal interface-services and secondary/ tertiary care settings such as emergency, orthopaedic, neurosurgery and pain management services [15–17]. Recent guidelines for the assessment and treatment of LBP and sciatica published by the National Institute for Health and Care Excellence (NICE) [6] call for a holistic approach to the management of LBP [7]. Owing to their skills in musculoskeletal assessment, including the appropriate ordering of diagnostic imaging and blood tests, physical therapies, management of psychosocial components of health and pain and appropriate listing of patients for spinal injection therapy, denervation and surgery, APPs are well placed to manage LBP in primary care [18, 19].

It is predicted that independent non-medical prescribing (iNMP) will be a core skill utilised by APPs working as FCPs, as the NICE guidelines for LBP recommend the use of analgesia within a multimodal management plan [6, 7]. A high quality mixed-methods systematic review evaluating the barriers and facilitators of iNMP acknowledged that the implementation of iNMP in contemporary clinical pathways may be successful if adequate preparation in terms of clinical governance, service and policy development and support for the clinicians are established prior to implementation [20]. Physiotherapist independent prescribers in the UK, have completed a post-registration iNMP programme and are regulated as 'independent prescribers' by the Health and Care Professions Council (HCPC). They are able to prescribe, administer or direct the administration of any medication (including those unlicensed), within their individual competence, scope and expertise, for any healthcare problem. Physiotherapist independent prescribing, although in its infancy, has been shown to have a good safety record and excellent patient satisfaction [21]. A recent systematic review evaluating the clinical and cost-effectiveness of iNMP across all professions internationally, identified limited evidence with unclear risk of bias. The systematic review demonstrates that no trials have evaluated the clinical and cost-effectiveness of physiotherapist independent prescribing in the context of LBP [22].

A stepped-wedge cluster randomised controlled trial (SWcRCT) design is proposed for use in a definitive trial owing to the contemporary nature of both the implementation of independent physiotherapy prescribing and the utilisation of APPs working as FCPs [23–25]. This research design allows for the use of fewer clinicians than those required for a parallel design. It is useful in the evaluation of the implementation of new interventions, being more reflective of current practice [24–27]. Although selection bias is considered a risk in cluster trials, the design is valuable when evaluating innovative clinical interventions where there is a strong ethical belief that the intervention will benefit patients [24, 26, 27]. The use of core outcome measures for LBP assessing pain intensity, health related quality of life and physical function are established in the literature [28]. To date, there is no agreement regarding the 'gold standard measures' for each of these outcomes. Patient reported outcome measures are frequently used for assessing pain intensity, health related quality of life and some aspects of functional activity. Quality systematic reviews have revealed that the physical activity of people with LBP is lower or equal to the healthy population [29–31]. The use of accelerometers to collect physical activity and sedentary behaviour data is advised in the literature, however although the feasibility of use with patients with chronic LBP has been established [32], the feasibility of fitting accelerometers for LBP in the context of an FCP clinic has not been evaluated to date.

A feasibility trial is required to inform the design of a definitive, low risk of bias, adequately powered, multi-centre SWcRCT investigating physiotherapist independent prescribing by APPs for patients with LBP in primary care.

## Aim and objectives

The aim and objectives of the feasibility trial are detailed in Table 1.

# Methods

This trial is approved by the Health Research Authority (HRA), ethical approval was sought via the Integrated Research Application System (IRAS) ID: 250734.

To ensure transparency and reproducibility, the feasibility trial was registered on the ISRCTN database (ISRCTN15516596a- registered 11[th] September 2018) and a detailed protocol was published [37] (S1 File). The authors confirm that all ongoing and related trials for this intervention are registered. The feasibility trial is reported in line with the CONSORT 2010 statement: extension to randomised pilot and feasibility trials. [38–40] (S2 Checklist). Patient and public involvement (PPI) is reported in line with the GRIPP2 short form reporting check list [41, 42].

## Ethics approval and consent to participate

To ensure that the trial was conducted in an ethical manner within best research practice, ethical approval obtained on the 30[th] October 2018 (IRAS project ID: 250734, Protocol number: RG_18–101, REC reference: 18/LO/1793) and HRA approval obtained, with R&D obtained from all sites [43, 44].

**Table 1. Aim and objectives.**

| Aim |
| --- |
| To evaluate the feasibility, suitability and acceptability of assessing the effectiveness of independent prescribing by APPs for patients with LBP in primary care to inform the design of a future definitive stepped-wedged cluster trial. |
| **General Objectives** |
| To assess the feasibility, suitability and acceptability of the proposed full trial [33] including: |
| • Eligibility criteria [34–36] |
| • Recruitment strategy [34–36] |
| • Data collection methods [34–36] |
| • Follow up procedures [34, 35] |
| **Specific Objectives** |
| *Feasibility*: |
| • To evaluate participant recruitment rates [33–35] |
| • To evaluate the ease of fitting participants with accelerometers and ease of data collection [34, 35] |
| • To evaluate the capacity (time and effort) of clinicians and researchers to complete trial related tasks [34, 35] |
| • To evaluate the necessary training requirements required by clinicians to successfully implement a full trial [34, 35] |
| |
| *Suitability*: |
| • To evaluate the range of participants' scores on the Roland and Morris Disability Questionnaire (RMDQ), assessing for floor effects and therefore the appropriateness of outcome measure for use in a full trial [33–36] |
| • To evaluate participant compliance with wearing the accelerometer device.[34, 35] |
| • To evaluate the time required to conduct each stage of the protocol [34, 35] |
| • To evaluate the appropriateness and availability of services and infrastructure such as access to national and institutional communication and information technologies required to undertake a full trial [34, 35] |
| |
| *Acceptability*: |
| • To evaluate the acceptability of the intervention to patients and the public [33–36] |

## Design

It is proposed that a SWcRCT design will be used to evaluate the clinical and cost-effectiveness of physiotherapist prescribing for LBP in the future. As the use of physiotherapist independent prescribing and the utilisation of FCP roles are new innovations in primary care, there are a limited number of APPs working as FCPs that are currently registered independent prescribers. Use of a stepped-wedge design allows APPs to cross from the control group to the experimental group once they are registered independent prescribers and start to utilise physiotherapists independent prescribing in their practice. This transition facilitates a robust and timely evaluation, which is reflective of current clinical practice. A full explanation of the use of the SWcRCT design for a definitive trial is detailed in the published protocol [37].

No existing framework describes best practice for completing feasibility trials in preparation for SWcRCTs [45]. Two-arm feasibility studies aiming to calculate intra-cluster correlation coefficients (ICCs) required for sample size calculations have been shown to exhibit insufficient accuracy [45]. Therefore, a prospective, mixed-methods, single-arm feasibility trial, exclusive of sample size estimation was employed to evaluate the trial objectives on the experimental arm of the future SWcRCT [34, 44, 46].

The mixed-methods approach comprised two component phases [47–49]:

1. Trial Component: a quantitative one-arm feasibility trial

2. Embedded Qualitative Component: qualitative semi-structured interviews and patient focus groups, using thematic analysis

## Trial component

A single-arm feasibility trial design was used to evaluate the trial objectives [36, 46]. Patient reported outcome measures (S2 File) were completed digitally via an online survey at initial assessment (baseline), 6 and 12 weeks following recruitment, to enable the evaluation of the data collection tool and the feasibility of follow-up data collection (date range for participant recruitment and final follow-up: Rural town 3rd December 2018-11th January 2019, final follow-up 15th April 2019; Regional city 28th November- 19th December 2018, final follow-up 13th March 2019; Capital city 28th February-18th July 2019, final follow-up 10th October 2019) [44, 50]. Follow up time points were selected based on prognostic literature demonstrating the 'normal' resolution time of LBP [51–53]. To allow real-time data capture and storage, the online outcome measures survey was built using REDCap (Research Electronic Data Capture) software hosted at the Centre for Precision Rehabilitation for Spinal Pain (CPR Spine) at the University of Birmingham, UK [54]. The number of participants that declined to participate as they were unable to complete the outcome measures survey online were collated to evaluate the suitability of only using digital data collection in a full trial.

Baseline measurements were collected in the clinical setting. At 6 and 12 weeks a link to the online survey was emailed to participants for completion. A reminder email was sent 24hrs and 48hrs later to facilitate compliance if a participant failed to complete the survey on the required day [44, 55]. To evaluate the feasibility of fitting participants with accelerometers in clinic, ease of data collection and participant compliance with wearing the accelerometer device for a 7 day period, the ten participants recruited at the rural town site had an accelerometer fitted to their left thigh for 7 days following the first consultation [34, 35]. Stamped/addressed envelopes were provided to enable return of the devices after use.

**Table 2. Participant inclusion criteria [37].**

| Inclusion Criteria |
| --- |
| • Male and female patients, aged >18 years. |
| • Non-specific LBP +/- leg pain requiring medication advice and drug prescription on assessment |
| • Classified as Moderate risk using the STarT Back Tool (classified as potentially benefiting from medicines and active physiotherapy treatment [5]) |
| • Able to read/communicate in English (owing to funding restrictions for interpreters and translators) |
| • Capable of following the demands inherent of the study |

| Exclusion Criteria |
| --- |
| • Signs of lumbar nerve root compression [61] |
| • Red Flags including potential spinal fracture, inflammatory disease, infection or malignancy [61] |
| • Spinal stenosis [62] |
| • Suspicion of or confirmed corda equine syndrome [63] |
| • Does not have capacity to consent [64] |
| • Unable to receive email and/or complete online questionnaires |

## Participants

The STarT Back Tool was used at initial assessment by the APPs to identify patients stratified into the medium risk LBP group [5]. This group of patients have been acknowledged as the prevalent group presenting for management of LBP in primary care; exhibiting both physical and psychosocial prognostic factors and potentially requiring physiotherapist prescribing to optimise their treatment outcomes [5, 56–58]. Patients in this group were eligible for recruitment if they met the inclusion criteria detailed in Table 2. Convenience sampling was employed as feasibility trials demand fluid recruitment and follow up with good participant retention [35, 44, 50, 59]. Convenience sampling was used as this method has the advantages of fluid recruitment. To minimise selection bias, patients fitting the eligibility criteria were recruited consecutively [60]. Patients interested in participating were provided with a participant information sheet (S3 File) explaining the rationale, content and research dissemination plans. Inclusion within the trial was entirely voluntary, with no incentives offered to participants to minimise bias [44, 50]. The APP answered patient queries and contact details for the research team were provided if the APP was unable to answer specific questions. Consent was obtained from willing participants using an online consent form (S4 File). Participants were free to withdraw at any time, without any impact on their care [44, 50].

## Interventions

The experimental arm of the definitive planned trial was used to evaluate the feasibility trial objectives [33–36]. An APP completed the initial assessment and management of participants in line with evidence-based practice. If medicines advice or prescription drugs were required/no longer required, these were prescribed/de-prescribed by the APP immediately.

## Outcomes

Outcome measures selected for use within the trial were informed by the literature and a team of subject-experts (including physiotherapists, pharmacists, medical practitioners, academics and health-service managers) and deemed most appropriate to evaluate the trial's objectives whilst attempting to minimise the burden on participants [28]. Detail of the primary and secondary outcome measures are detailed in Table 3. Assessment of the quality of sleep via accelerometer data was detailed in the published protocol [37]. Unfortunately, the devices available

**Table 3. Secondary outcome measures and their rationale [37].**

| Outcome | Measure | Rationale |
|---|---|---|
| **Primary Outcome Measures** | | |
| Pain | Numerical Rating Scale (NRS) | The NRS is a unidimensional 11-point scale (0–10) used to measure pain intensity, where 0 represents no pain and 10 represents maximum pain (e.g. the worse pain you can possibly imagine). [66] Patients with pain have been shown to prefer the NRS over other pain measure including the pain Visual Analogue Scale (VAS) owing to simplicity and clarity.[66, 67] The NRS has demonstrated good reliability, validity and responsiveness and has been used extensively in pain research.[68–70] A reduction of 2.5 points on the NRS has been shown to be clinically important for chronic LBP.[69–71] Participants scored pain in 3 categories: "worst pain over the last two weeks", "least pain over the last two weeks" and "average pain level today". |
| Disability | Roland Morris Disability Questionnaire (RMDQ) | The RMDQ is one of the most widely used outcome measures for LBP, with well-established good levels of validity and reliability.[72] The RMDQ has been selected over its counterparts owing to its superior measurement properties in patients reporting moderate disability demonstrated by those stratified into the medium risk group by the STarT Back Tool.[5, 71, 72] The 24-item questionnaire takes approximately 5 minutes to complete and includes items assessing: physical activity, sleep, psychosocial factors, activities of daily living, appetite and pain.[73] Scores range from 0 (no disability) to 24 (maximum disability), with a change of 3.5 points deemed clinically significant.[71] |
| **Secondary Outcome Measures** | | |
| Health Related Quality of Life (QALY) | EQ-5D 5L | The EQ-5D 5L is used to measure health related quality of life demonstrating good reliability and validity through psychometric testing [74]. If feasibility is found this measure will inform cost utility in a full RCT. |
| Pain Related Fear of Movement | The Tampa Scale for Kinesiophobia (TSK) | The Tampa Scale for Kinesiophobia (TSK) is a 17-item tool which was developed to measure a person's fear of movement owing to LBP. Ongoing fear of movement has been linked to the development of long term persistent pain [75]. This outcome measure has been found to show good validity and reliability when measuring pain related fear of movement [76]. |
| Physical activity and | ActivPal 3 Accelerometer | Anecdotal evidence suggests that decreasing sedentary behaviour in people with LBP may have significant health benefits [57], reducing risks of obesity, metabolic syndrome, type two diabetes and mortality [77]. Systematic reviews have revealed that physical activity of people with LBP is lower or equal to the healthy population [29–31], however there appears to be differing patterns of physical behaviour, with the back-pain population engaging in shorter bouts of physical activity which are not long enough to incur health benefits (>10 minutes) [31, 78]. An accelerometer will be used to collect data including: time sitting, standing and walking, steps count and overall activity score [65]. Where necessary, participant diaries mapping activity, sedentary time and time asleep can be used to differentiate between sedentary time and sleep [65]. To date no individual brand/model of accelerometer has been identified as gold standard. The ActivPal 3 was selected for use in this feasibility trial as it has been seen to be more precise and sensitive than other accelerometers [65, 79]. |
| Time to return to work and nature of return to work (e.g. full time, part time, light duties) | Days | Work absence owing to sick leave for work disability is a key issue clinically, socially and economically. The MCIC for time return to work has not been defined owing to the specific measurement (days on sick leave) being widely accepted and recognition of the measure's value in social and economic issues rather than an indicator of morbidity [71]. This measure would therefore be useful when conducting economic evaluation of physiotherapist prescribing. |
| Prescription Utilisation, Participant | Days | Time requiring drugs for the treatment of non-specific LBP discussed/prescribed by the advanced physiotherapists was monitored to evaluate the necessity of this measure for future cost-effectiveness analysis within a full trial. |
| Number of appointments with other healthcare professionals about this episode of LBP | Number of appointments with each type of healthcare professional | The number of appointments with other healthcare professionals about the specific episode of LBP being studied was recorded via a question in the outcome questionnaire to evaluate the necessity of this measure for future cost-effectiveness analysis within a full trial. |

for testing the feasibility of fitting the accelerometers in FCP clinics and evaluating participant's tolerance of wearing the devises, were not validated for measuring quality of sleep. Previous quality research has established that it is feasible to evaluate time sleeping using the data collected by the devises when cross referenced with a participant diary mapping activity, sedentary time and time asleep [32, 65]. It was not deemed necessary to re-evaluate this process. This was the only deviation from the feasibility trial protocol.

## Sample size

Three APPs, across 3 primary care sites representative of English geography, recruited up to n = 10 participants each within a 6 month recruitment period. This enabled evaluation of recruitment rates across clinicians and the feasibility of the trial methods in both metropolitan and rural healthcare services [34, 45, 46]. A sample size of n>20 is regarded as adequate within the literature, when testing feasibility objectives for cRCTs, however a total sample of n = 30 participants was planned to allow for under-recruitment within the specified time period and loss to follow up [34, 35, 45, 46].

## Data analysis

Participant flow and loss to follow up was described using a CONSORT diagram to evaluate the feasibility of eligibility criteria and acceptability of recruitment and follow up rates [38]. Data from fully completed outcome questionnaires were included in the data analysis. Data were tabulated, and primary descriptive analysis was completed to test procedure [34, 44, 50]. Effectiveness was not statistically analysed as this was not within the scope of the feasibility trial [44, 50]. As a definitive trial will aim to evaluate the clinical effectiveness of independent prescribing by APPs for patients with LBP in primary care, data distribution of the primary outcome measures across participants, was evaluated at baseline, 6 and 12 weeks, with 12 week data used to measure for a potential floor effect [80].

## Embedded qualitative component

**Design.** For clarity and transparency, the qualitative component is reported using the Consolidated Criteria for Reporting Qualitative Health Research (COREQ)[81]. Qualitative methods aimed to assess the APP and patient participants' views, perceptions and experiences related specifically to the trial objectives [33–36, 82, 83].

**Advanced physiotherapy practitioners.** APPs were evaluated via semi-structured in-depth face to face interviews, undertaken by one researcher (TN) following completion of participant data collection. Question design was informed by the methodological literature and developed by a team of experts in the fields of physiotherapy, primary care, non-medical prescribing (NMP), health policy and trial methodology [44, 59], then reviewed for clarity and appropriateness by a patient and public involvement (PPI) group (S5 File) [84]. Prior to interview, consent to participate was gained following the provision of a participant information sheet and responding to questions. Interviews were recorded and transcribed verbatim. To ensure all views and were captured, transcripts were reviewed by participants for comments and amendments prior to analysis [82].

**Patients.** Patient data were collected via a focus group following the 12 week assessment point [34, 85], as this method is recognised to produce rich data representative of a collective view point [86]. A purposive sample of 6 patients (representative of ages and gender) was utilised, as this sample size is reported as optimal in the literature [85]. The focus group was conducted by two researchers (facilitator and observer), using a predetermined topic guide (S6 File) developed by a team of experts and informed by the methodological literature [44, 59].

The topic guide was reviewed prior to use by a PPI group to ensure appropriateness and clarity [84]. Consent to participate was gained prior to the focus group commencing. The participants received a participant information leaflet and had the opportunity to have any questions answered. The focus group was recorded using a digital audio recording devise and transcribed verbatim. Transcripts were returned to participants for comments/correction to ensure all views were represented [81].

## Analysis and findings

A grounded theory theoretical framework enabled a thematic analytical approach to analyse and synthesise the qualitative data. This method enables identification of the important thoughts and views of the population being studied, providing explanations alluding to how the concerns may be resolved or processed in preparation for a full trial [44, 87, 88]. Transcripts were coded line-by-line using NVivo 11 software (QSR International, Melbourne, Australia) by one researcher (TN) and verified by a second researcher (AR) [50, 88, 89]. Rigorous comparative analysis was completed to identify similarities and differences within the data, informing the development of descriptive categories which were linked, merged or split to synthesise a conceptual understanding of the data [88, 89]. To avoid single researcher bias, the second researcher (AR) re-interrogated the data to validate or contradict findings. Following this process, to ensure trustworthiness, outcomes were discussed with a panel of experts for confirmation and agreement [87, 88, 90].

## Data storage

All data were stored in password protected computer files that could be accessed only by trial investigators at the University of Birmingham. The password-protected files will be retained for 10 years satisfying university code of practice.

## Integration: Feasibility, suitability and acceptability

Following quantitative and qualitative analysis, data were assessed against a priori defined success criterion developed by experts and informed by the methodological literature [44, 59, 91]. Success criteria can be found in S7 File. Trial objectives were considered successful if the success criteria were satisfied following the integration of the quantitative and qualitative findings [35, 91].

## Patient and public involvement (PPI)

Patients with LBP were part of the research team to ensure the patient perspective was central to planning and decision making. There was a PPI representative on both the trial management group and trial steering group to ensure that patients and the public were involved at all steps of the research process. Patients were involved in the development of the participant information sheet, consent form and questions used in the semi-structured interviews and focus group.

# Results

## Trial component

**Demographics and participant flow.** Demographic and recruitment data are presented in Table 4. n = 29 participants (n = 12 male, n = 17 female) were recruited. The mean recruitment rate was 1.07 participants/week. Two sites recruited the pre-defined n = 10 participants within the 6 month recruitment period (3 and 4.5 weeks). The capital city site recruited n = 9

**Table 4. Demographic and recruitment data.**

| Gender | | |
|---|---|---|
| Male | 12 | |
| Female | 17 | |
| Age | | |
| 17–29 | 3 | |
| 30–39 | 8 | |
| 40–49 | 5 | |
| 50–59 | 5 | |
| 60 or older | 8 | |
| Recruitment rates | | |
| Location | Time to recruit (weeks) | No of participants (n) |
| Rural town | 4.5 | 10 |
| Regional city | 3 | 10 |
| Capital city | 20 | 9 |
| | Mean (SD) = 9 (9.41) | Total = 29 |

participants over the 6 month period. Successful loss to follow up was defined a priori as <20%. 48% of participants were lost to follow up at 6 weeks, with 65.5% at 12 weeks (Fig 1). One site had a loss to follow up of 89%, suggestive of site-specific issues. No patients refused to participate owing to the inability to complete the outcomes measure survey online.

## Outcome measures survey

Table 5 presents mean primary and secondary outcome measure data collected from the outcome measure questionnaire with variability reported by the use of standard deviations (SD). Reductions in pain were found for all pain categories as time progressed. Mean scores on the RMDQ reduced from 9.21 (SD 5.58) at base line to 8.07 (SD 5.82) at six weeks, then increased to 9.70 (SD 5.33) at 12 weeks. Between baseline and 12week, improvements were seen across all components of the secondary outcome measures other than anxiety and depression (EQ-5D 5L) which increased with time. No participants scored the distinct lower limit in any of the outcome measures; therefore, no floor effects were found. As primary and secondary outcomes improved, absence from work and prescription utilisation reduced.

## Accelerometery

Ten participants (n = 2 male, n = 8 female) wore an ActivPal accelerometer 24 hours a day for seven days. Data collected by the accelerometers are displayed in Table 6. There were no missing data. Participants spent an average of 18.57hrs (SD = 1.54) sitting per day, 4.14hrs (SD = 1.17) standing and 1.3hrs (SD = 0.39) walking. Participants completed 5884.66 steps per day (SD = 2255.11), with a mean activity score of 32. 94MET.h (SD = 1.03).

## Embedded qualitative component

**Interviews: Advanced Physiotherapy Practitioners (APPs).** *Demographics.* Demographic details of the APPs can be found in Table 7. APPs all had a post-graduate qualification in the musculoskeletal specialty. The number of years qualified as a physiotherapist ranged from 15–28 (mean = 21 years).

The APPs' (n = 3) views, perceptions and experiences related specifically to the trial objectives were analysed and synthesised into three themes and associated subthemes:

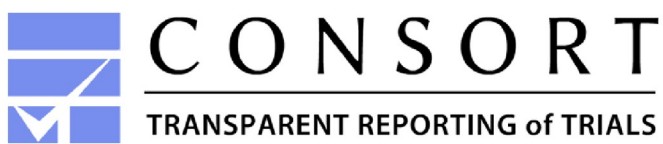

## CONSORT 2010 Flow Diagram

**Enrolment**

Fulfilled eligibility criteria
(n=36)

Excluded (n=7)

Declined to participate (n=7)

Randomised (n=0)

**Allocation**

Allocated to intervention (n=29)

Received allocated intervention (n=29)

Did not receive allocated intervention (n=0)

**6week follow-up**

Lost to follow-up (n=14)

Rural town (n=1), Regional city (n=6), Capital city (n=7)

**12week follow-up**

Lost to follow-up (n=19)

Rural town (n=4), Regional city (n=7), Capital city (n=8)

**Fig 1. CONSORT flow diagram [38–40].**

1. Trial design, conduct and processes

   a. Eligibility criteria, recruitment strategy and follow up procedures

   b. Capacity (time and effort required)

**Table 5. Primary and secondary outcome measures data.**

| Outcome measure | Baseline Mean (SD) | 6 weeks Mean (SD) | 12 weeks Mean (SD) |
|---|---|---|---|
| Pain | | | |
| Worst pain over the last 2weeks (0–100) | 81.17 (18.18) | 59.87 (27.84) | 58.2 (31.88) |
| Least pain over the last 2weeks (0–100) | 43.48 (25.72) | 34.07 (23.88) | 25.7 (20.01) |
| Average pain level today (0–100) | 55.89 (23.18) | 42.53 (23.61) | 40.4 (28.86) |
| RMDQ (0–24) | 9.21 (5.58) | 8.07 (5.82) | 9.7 (5.33) |
| EQ-5D 5L | | | |
| Mobility (0–5) | 2.45 (0.99) | 1.93 (0.96) | 2.2 (0.79) |
| Self-care (0–5) | 1.76 (0.83) | 1.53 (0.74) | 1.6 (0.84) |
| Usual activities (0–5) | 2.66 (1.01) | 2.40 (0.99) | 2.5 (0.85) |
| Pain/discomfort (0–5) | 3.24 (1.02) | 2.33 (0.72) | 2.6 (0.8) |
| Anxiety/depression (0–5) | 1.66 (0.72) | 1.80 (1.21) | 2.0 (1.33) |
| Health today (0–100) | 74.72 (27.18) | 68.2 (15.65) | 59.6 (21.37) |
| TSK-11 (11–44) | 25.66 (7.99) | 24.13 (8.64) | 22.2 (7.92) |
| Absent from work (DAYS- between each survey) | 8.52 (20.04) | 7.07 (17.38) | 1.0 (3.16) |
| Total prescription utilisation (DAYS) | - | 17.47 (17.3) | 25.3 (37.08) |
| | | | |
| Number of appointments with other health professionals (between each survey) | | | |
| General Practitioner | | 3 | 0 |
| Advanced Physiotherapy Practitioner | - | 1 | 1 |
| Spinal surgery team | | 1 | 0 |
| Pain Management team | | 0 | 1 |

**Table 6. Accelerometer data across 7days.**

| Participant | Sitting (hrs) | Standing (hrs) | Walking (hrs) | Steps | Sit-Stands | Activity Score (Metabolic Equivalents- hours (MET.h)) |
|---|---|---|---|---|---|---|
| 1 | 17.60 | 4.66 | 1.77 | 9720.86 | 49.57 | 34.47 |
| 2 | 18.13 | 4.36 | 1.53 | 6386.29 | 64.00 | 33.19 |
| 3 | 15.67 | 6.49 | 1.86 | 8680.29 | 55.71 | 34.38 |
| 4 | 19.67 | 3.43 | 0.90 | 3461.71 | 32.00 | 31.90 |
| 5 | 21.80 | 1.71 | 0.50 | 1808.29 | 33.57 | 30.98 |
| 6 | 18.34 | 4.31 | 1.36 | 5966.00 | 56.29 | 33.00 |
| 7 | 18.54 | 4.16 | 1.32 | 6003.90 | 48.52 | 32.99 |
| 8 | 18.54 | 4.16 | 1.32 | 6003.90 | 48.52 | 32.99 |
| 9 | 18.67 | 4.09 | 1.25 | 5472.91 | 48.37 | 32.77 |
| 10 | 18.75 | 4.05 | 1.22 | 5342.43 | 46.14 | 32.72 |
| Mean Total-over 7 days | 18.57 | 4.14 | 1.30 | 5884.66 | 48.27 | 32.94 |
| SD | 1.54 | 1.17 | 0.39 | 2255.11 | 9.74 | 1.03 |

**Table 7. Demographic data of focus group participants.**

| No: | Job title | Gender | Years registered as a physiotherapist | Post graduate qualifications |
|---|---|---|---|---|
| 1 | Advanced Physiotherapy Practitioner | Male | 15 | MSc Musculoskeletal Physiotherapy Non-medical Prescribing |
| 2 | Advanced Physiotherapy Practitioner | Female | 28 | MSc Musculoskeletal Medicine Non-medical Prescribing |
| 3 | Advanced Physiotherapy Practitioner | Female | 20 | MSc Musculoskeletal Physiotherapy Non-medical Prescribing |

 c. Training requirements

2. Data Collection, outcomes and measures

 a. Data collection tool and accelerometery

 b. Equipment, services and infrastructure

3. Adequacy of the feasibility trial

## 1. Trial design, conduct and processes

1. *Eligibility criteria, recruitment strategy and follow up procedures*
APPs felt that a full trial should enable the evaluation of all stratifications of NSLBP +/- leg pain. They advocated for a full trial that include participants across all LBP stratification groups defined by the STarT back tool. This would allow for both intra- and inter-group comparisons evaluating the effectiveness of independent prescribing by APPs for patients with LBP in primary care across the spectrum. The APPs advised that variation would exist regarding the utilisation of skills within the scope of independent prescribing across each of the STarT back stratification groups due to patient-specific factors such as the length of time the participant has suffered with LBP, associated psychosocial factors, previous treatments and experiences and related medical history. Therefore, it was felt that the specific prescribing skills used, such as prescribing medicines, de-prescribing, advising about the use of over-the-counter medication and medicines management, should be captured throughout the individual patient journey. All APPs expressed that the trial methods tested by the feasibility trial would enable appropriate comparison of STarT back stratification groups, however they highlighted that the complex nature of persistent LBP would require follow-up for longer than the 12-week period tested, demonstrating the need to update the primary end point for a full trial.

All the APPs felt that the recruitment strategy was suitable, although recognised that recruitment rates could potentially differ between APPs in a full trial due to varying experience of clinicians working in FCP roles and proficiency and/or confidence in prescribing medicines. Two of the clinical sites (regional city and rural town) recruited n = 10 participants over 3- and 4.5-week periods respectively. The third site (capital city) used the full six-month recruitment period to recruit n = 9 participants. The APP from the capital city site reported that patients were frequently excluded from participation due to language barriers. Additionally, English-speaking young professionals in the area often declined recruitment, declaring that they were too busy to commit to complete the 6 and 12-week follow-up surveys. Recruitment was curtailed across all sites by patients consistently stating that they did not want to take analgesia for their LBP, preferring to engage in other conservative management strategies such as exercise and manual therapies. This was cited as the key benefit of consulting a physiotherapist for their LBP rather than their GP.

To optimise recruitment rates in a definitive trial, it was suggested that posters and social media posts might encourage patient interest. Administrative staff could be utilised to highlight potential participants when booking appointments and provide patients with participant information leaflets prior to seeing the APP. APPs across both city-based sites recognised the risk associated with participants using data collection equipment unsupervised, fearing that there was potential for theft or damage. Patient's literacy was highlighted as a potential barrier to independent completion of the online outcome measures surveys, and thus a barrier to recruitment of a sample representative of the LBP population. All

APPs highlighted that due to restricted consultation time in FCP practice, the use of research assistants to recruit and consent participants identified by the APP during consultations, could further optimise recruitment rates. APPs postulated that research assistants at each site, recruiting, consenting and aiding patients with completing the online outcome measures survey, would not only simplify recruitment and follow up procedures but would also minimise risk, especially at locations where multiple APPs are recruiting to a full trial. The APPs advised that to improve compliance with follow-up procedures, participants should be asked to consent to reminder telephone calls from the research team, with the choice of completing the follow-up outcome measures survey online, on paper, independently or face-to-face with a researcher or research assistant. It was proposed that these changes would increase the likelihood that patients would consent to participating in the trial and minimise drop-out due to non-compliance.

2. *Capacity (time and effort required)*
Twenty-minute FCP consultations were scheduled at each site. The APPs stated that the recruitment and consent of each participant and completion of the initial outcome measures survey took approximately 10–15 minutes. If the application of an accelerometer was required, this took an additional 5–10 minutes dependant on the "tech savvy nature" of the individual participant. Additional time was also required to answer participants' questions, confirm data entry and upload data to the REDCap server. All APPs described the time pressures as "stressful", causing their clinics to run late. Further, all APPs reported "rushing" clinical assessments if they were aware that they needed to recruit a patient to the trial. It was recommended that if APPs are to be used to recruit, consent, aid in the completion of the initial outcome measures survey and apply an accelerometer in a full trial, 30-40-minute consultations would be necessary. Unfortunately, all of the APPs agreed that healthcare service commissioners would not agree to extended appointment times to allow for research activity. Therefore, a full trial would require a research assistant at each site to ensure appropriate clinical and administrative capacity.

3. *Training requirements*
All APPs reported that they felt prepared and confident to lead the recruitment of participants and initial data collection processes. Each APP reported acting as principal investigators for previous clinical trials and observed that less experienced clinicians may require additional training in ethical recruitment and consent procedures, documentation and data management. All recruiting clinicians would require training to effectively apply accelerometers to participants and to educate participants on the re-application should dressings become unsealed or cause a skin reaction. For future trials it was recommended that peer support and clinical mentorship would be essential for new prescribers and less experienced APPs. This would ensure best practice and assist in managing variation in clinicians' confidence to prescribe.

## 2. Data collection, outcomes and measures

1. *Data collection tool and accelerometery*
All of the APPs agreed that the outcome measures within the survey were suitable and encompassed the majority of the multi-faceted dimensions of LBP. Two of the APPs recommended that tools evaluating sleep, lifestyle and confidence with global physical activity could further improve the richness of data collected in a full trial. Although the APPs recognised that the outcome measures within the survey were validated and tested for reliability

and reproducibility, it was reported that some of the participants required help to complete all questions in the survey. It was suggested that this variation in understanding was due to differing participant literacy levels.

The survey logic was reported as user-friendly other than an issue with one question ("Number of days taken off work?") as this question was not applicable to participants who were part-time, retired or registered as disabled. The APPs all commented that the difficulty in the ability to capture data about the wider use of the independent prescribing qualification was a weakness. The use of prescribing, de-prescribing and medicines advice and management were deemed fundamental to the effective use of NMP within the FCP role. APPs were also interested in capturing data around what, as well as how drugs are currently being prescribed for the management of LBP by APPs in primary care across the country. No problems were reported regarding the fitting of accelerometers. Participants were happy to be taught to re-apply accelerometers themselves should the dressings become loose or need replacing. No issues were reported to the APPs regarding self-removal of accelerometers by participants after seven days, and all units were returned in a timely manner without damage. No adverse effects or compliance issues were reported. It was highlighted that all participants lived locally to the clinic and returned the accelerometers in person. If participants lived further away from clinical sites it was deemed acceptable for participants to return accelerometers by post following provision of a padded pre-paid envelope.

2. *Equipment, services and infrastructure*

    Overall, the APPs agreed that the services and infrastructure tested by the feasibility trial were suitable and would enable the completion of a robust full trial. All of the APPs disclosed technical issues with the tablet computer provided for data collection. One APP stated that the tablet was not sophisticated enough to optimally run the REDCap application. The others described the application as "clunky", reporting significant issues with the vertical sliding visual analogue scale (VAS) within the EQ-5D 5L tool. The security of the tablet computer was a concern if the outcome measures survey was to be used outside of the consultation room. It was recommended that the tablet computer should be securely attached to a wall in a quiet waiting area to stop theft, however it was accepted that participant privacy and dignity may preclude this solution from being a valid alternative.

### 3. Adequacy of the feasibility trial

Each of the APPs interviewed stated that the feasibility trial was adequate for assessing the feasibility suitability and acceptability of the trial methods. The clinicians all agreed that iNMP enables better holistic care by FCPs for LBP in primary care, therefore a full trial is essential to evaluate efficacy. The APPs deemed all ethical conduct and trial documentation (including participant information sheet, consent form) acceptable for use in a full trial. The use of digital consent and data collection was seen to be positive, securely storing participant data and enabling the blinding of the analytical research team.

To better inform clinical decision making, the clinicians enquired as to whether collaboration with NHS digital might further streamline data collection. It was felt that connectivity between data collection technology and the participant's digital clinical notes, might highlight psychosocial factors not identified by clinicians in a time-pressured consultation, further improving management of the 'whole patient'.

Table 8 provides illustrative quotations to demonstrate each theme.

**Table 8. Interviews, comments that reported or discussed each theme and illustrative quotations from APPs (quotations have been copied verbatim).**

| Theme | Illustrative Quotations |
|---|---|
| Trial design, conduct and processes | "I think it's quite time intensive with the clinician doing it [recruitment] in real terms, mid-clinic, to collect all of that data." (APP1) |
| | "The reasons it [recruitment] was slow was, I didn't prescribe much, and I think the biggest reason was because patients didn't want to take medication for their back pain. …..." (APP 3) |
| | "rather than having 20-minute slots for these patients, I would want a good 30–40 minutes for them, so that I didn't rush those elements [clinical assessment]." (APP 1) |
| | "The ideal situation, I think, would be for the first contact practitioner to identify the patient and then create a list for a research assistant to then take over, to put the bits and bobs on and to sit with them to do the questionnaire." (APP2) |
| | "….. most of them kind of knew how to use a tablet, but I definitely couldn't just ignore them to let them get on with it in the waiting room…" (APP 1) |
| | "You definitely need a research assistant. If you're going to sit with somebody, then I think the clinician doesn't really have enough time to do that." (APP 2) |
| | "Time was massive, time, restricted anyway with 20 minutes, appointments, and I felt that it did cause me to run over." (APP 3) |
| | "…. in the demographics, with all due respect, that we were in, I didn't really want to give them a brand-new tablet to take away and do it because there was the potential that that wouldn't come back." (APP 1) |
| | "… there were 10 subjects, on average round about five minutes I would say. Some were a bit less, there was one or two patients who were a good 10 minutes or so, really, who weren't tech savvy." (APP 1) |
| | "Time factor, they [patients] didn't want to [be recruited] … they were maybe happy to do the first questionnaire, but they didn't want to then do the follow-up questionnaires, they couldn't commit to it." (APP 3) |
| | "A way round it [time restrictions] would have been to recruit an admin staff or an assistant and for them to go through it in person with those patients, the questions, and their input." (APP1) |
| | "I think having someone there with them, not to bias the answers but to just read it along with them, was useful." (APP 1) |
| Data Collection, outcomes and measures | "… these are standardised questionnaires and they're robust and they've been well studied, but they still did ask questions and they were not 100% sure of what they should do." (APP 1) |
| | "I thought the content was good, there wasn't anything particularly in there that I thought shouldn't have been in there." (APP 2) |
| | "It felt like it was set up on an Apple and then put on an Android because it was clunky software." (APP 1) |
| | "I think the questions that are on there are all well-studied, reliable, robust measurement tools, as robust as we can have." (APP 3) |
| | "I thought the content was good, there wasn't anything particularly in there that I thought shouldn't have been in there." (APP 2) |
| | "… if you think about back pain and all the contributing factors. …. so, lifestyle and sleep [should be added]." (APP 3) |
| | "I think the accelerometers are really good at doing what they're supposed to do in terms of activity, lying down, standing up. It would be quite interesting to correlate that to actual activity." (APP 2) |
| | "I think the benefits of electronic and automatic. …. getting it [surveys] uploaded. … would be good, I think that would be preferential, over paper." (APP 1) |
| | "I quite like the slidey things, but the vertical slidey thing doesn't work on an iPad, it just moves the whole pad up and down" (APP 2) |
| | "Yeah, the age demographic didn't play out necessarily. Some of my older patients could fly through it." (APP 1) |
| | "I'd be keen for you to collect information about what drugs were we looking at, what exact prescribing decisions we would be making." (APP1) |
| | "I liked it being electronic. I thought, having the patient information sheet in paper was quite good because you could go through that together. But I thought in terms of the actual rest of it, I thought it was fine." (APP 3) |
| | "They [patients] were given prescription advice, or de-prescribed… that was the bulk of my work… And getting them to take medication correctly." (APP 1) |
| | "There was one particular section that I needed to explain to patients, it was the bit where they were looking at patient statements of pain." (APP2) |
| | "You've definitely got different categories and yeah, you've got those ones… it's quite different the ones you've written the prescription to, to those ones that are just over the counter. Or GP has given you this, but actually you're not quite taking it right. That would reflect what we really do." (APP 3) |

*(Continued)*

**Table 8.** (Continued)

| Theme | Illustrative Quotations |
|---|---|
| Adequacy of feasibility trial | ". . .. the biggest part of it, it's about advising about what pain is and about how it can be managed and it's not dangerous and about how activity is more beneficial to them than not being active. It's about rehabilitating, psychologically and physically back to full function." (APP 2) |
| | ". . . people that come and they're on opiates when they don't need to, beforehand, if you weren't qualified in prescribing, you might not know that they're inappropriate and you might not have the confidence to venture into that and challenge that prescription decision. That patient would continue to take a drug that might be doing more harm for them than benefit, which is not great." (APP 1) |
| | "A minimal amount of my prescribing might be in the acute things, episodes where they might need a prescribed drug, but you can then quickly bring them off that drug. Also, more importantly, is the fact that when people are put onto, say opiates, for example, an FCP is more likely to take the person off the opiates, by de-prescribing or they can reduce the pain medications down in a graded kind of way to make sure they're safe and then go onto over the counter drugs." (APP 2) |
| | "I thought the information sheet was really thorough and I thought it was clear and the patients seemed to understand it." (APP 1) |
| | "I liked it being electronic. I thought having the patient information sheet in paper was quite good because you could go through that together." (APP 3) |
| | "I think the questions that are on there are all well-studied, reliable, robust measurement tools, as robust as we can have." (APP1) |
| | "I do think it'll work; I think it's good but again, the de-prescribing part needs to be in it." (APP 2) |
| | "it does cover kinesiophobia and fear and those elements as well as ADL, so I think it's a rounded array." (APP1) |
| | "It's about rehabilitating, psychologically and physically back to full function. And pain management can be hot or cold. It doesn't have to be as in medication although medication is part of the whole thing and it is used, but I think that that's a small part of the patient's recovery for their back pain." (APP 2) |
| | "Being able to de-prescribe I think has been probably the most beneficial part of doing that role and then prescribing, you're enabling patients to get on board with their treatment and therapy." (APP 1) |
| | "Overall, I think it's very beneficial to have those skills and the ability to tap into them is really useful. It increases my own self-confidence when exploring drug histories and putting the bigger picture into place. Rather than having a niche of physio and not daring venture into medication because of medico-legal processes, I'm happy to stray into those topics. I think the patients holistic care is better for that." (APP 1) |
| | "I wouldn't necessarily put the patient on pain medication anyway. I'd be more likely to be advising activity, movement and explain the pain is not dangerous and stuff like that, rather than putting them onto a medication." (APP 2) |

## Focus group: Patients

**Demographics.** Focus group demographic data is presented in Table 9. Six participants from the trial component of the feasibility trial consented to participate in the focus group. Purposive sampling enabled a representative spread of ages. 66% (n = 4) of the participants were female, 66% (n = 4) of participants completed the feasibility trial, with 33% (n = 2) lost to follow up at 12 weeks.

**Table 9. Demographic data of focus group participants.**

| Demographic Descriptor | Number of Participants |
|---|---|
| Male | 2 |
| Female | 4 |
| Age (years) | |
| 17–29 | 1 |
| 30–39 | 1 |
| 40–49 | 1 |
| 50–59 | 1 |
| 60 or older | 2 |
| | Total number of participants n = 6 |

Patients' views, perceptions and experiences related specifically to the trial objectives were analysed and synthesised. Data were synthesised into three themes:

1. The use of physiotherapist independent prescribing in FCP clinics

2. Trial conduct and processes

    a. Recruitment processes

    b. Follow up processes

3. Data Collection

    a. Accelerometery

    b. Outcome measure survey

## 1. The use of physiotherapist independent prescribing by APPs

The participants expressed that they were happy with the introduction of physiotherapist independent prescribing and felt confident in the APPs' skills. All participants agreed that physiotherapists are experts in the management of LBP. APPs were felt to provide a more detailed assessment than GPs, listening to patients' problems prior to developing a holistic treatment plan alongside the individual patient. All participants agreed that they would prefer to utilise non-pharmacotherapy methods to manage their LBP. If analgesia was required by a patient to facilitate management and rehabilitation, the APP was able to advise why the medication was needed and how best/ when to take the medication within the context of their social and family life, work commitments and associated treatments, including exercise therapy and physical activity. It was recommended by participants that more clinicians in similar roles should be employed across the whole spinal pain pathway, particularly within urgent care centres and emergency departments.

## 2. Trial conduct and processes

1. *Recruitment processes*
   The participants reported that they were happy with the ethical conduct throughout the feasibility trial. They deemed the participant information sheet to be satisfactory and suitable for use in a full trial. All participants were happy with the recruitment process and could not identify any adverse effects or risks to being involved in the trial. It was proposed that the use of posters, social media and advertising on waiting room television screens might encourage participation in a full trial. Further, the focus group recommended marketing aimed to involve all NHS patients in research activity. It was thought that this broader marketing strategy would reduce fear of participation and increase public awareness about the social responsibility for participation in health research.

2. *Follow-up procedures*
   All participants agreed that the follow-up procedure used in the feasibility trial was acceptable however, participants in a full trial would benefit from choice of communication options. Participants agreed that personal preference would vary between patients, some preferring contact via telephone or post rather than email. They requested that each clinical site should have a liaison point for face-to-face discussion if required, with the option to complete the follow-up outcome measure surveys digitally or on paper, over the telephone,

via video call or face-to-face with a member of the research team. These options should be offered to the participants during the consent procedure with the participants being able to change their preference during participation in the trial if required. It was posed that the probable reason for non-response to the six week or 12-week follow up survey was that the participants were no longer suffering from LBP. It was felt that the clinicians recruiting participants to a full trial should be explicit about the necessity to complete all follow-up surveys whether LBP had resolved or remained present. It was felt that reminders from a research assistant would further assist compliance.

## 3. Data collection

1. *Accelerometery*
   Participants agreed that all aspects of the feasibility trial testing the use of accelerometery suitable and acceptable. They reported that the APP's explanation regarding the application, reapplication and rationale for use of the accelerometer was clear and understandable. Participants felt confident to reapply the accelerometer with a fresh dressing if required. However, this was not necessary across the trial participants. Three participants reported worrying about re-positioning the devices if re-application was necessary, however were happy that they could seek help from their APP if required. It was recommended that a further patient information sheet should be developed demonstrating the use, application and removal of the accelerometer to prompt a participant's memories.
   Participants reported that the devices were easily fitted, and none experienced any adverse effects. One participant reported slight skin irritation in the final hours wearing the device but did not feel this was bad enough to warrant a change of dressing. Another participant stated that although wary of her sensitive skin during application, she had no reaction to the device-cover or adhesive dressing. Participants all concurred that the use of accelerometers attached to the skin enabled them to forget that they were wearing the device and continue with normal activity. It was felt that the device did not prompt additional activity after the first 24 hours and were no problem during land- or water-based exercise. Participants reported that attaching the accelerometer to the skin was preferable to wearing a device around the wrist, ankle or on their clothing, as these types of devices might prompt additional physical activity that they would not have otherwise undertaken. It was felt that removable accelerometers might provoke feelings of stress due to a sense of constant examination and worry that results would be skewed if the device was removed and not replaced immediately.

2. *Outcome measure survey*
   All participants agreed that the outcome measures survey was suitable for assessing the progression of their LBP. Completing the initial survey on a tablet was acceptable as long as help was available from a researcher or research assistant if required. The participants describe the tablet as "clunky", explaining the problem with using the vertical sliding scale within the EQ-5D 5L tool.

Some participants reported difficulty in understanding the wording contained within the Tampa Scale for Kinesiophobia, other than this all other questions within the outcome measures survey where deemed clear and understandable. Participants debated the use of a 10-point or 100-point NRS, however no consensus in preference was reached, concluding that both numerical scales are acceptable. They warned that participant's answers may vary dependent upon the time at which the survey is completed relative to a patient's diurnal pain pattern

and the timing of analgesia. However, participants also acknowledging that dictating a specific time for survey completion would not be feasible due to variation in participant's daily lives. Overall, the participants agreed that the survey evaluated their LBP journey well but recommended the formal assessment of sleep within a full trail.

Table 10. Provides illustrative quotations to demonstrate each theme.

### Integration: Feasibility, suitability and acceptability

For the 'general' trial objectives 90% of the success criteria were met. Both the general and specific objectives demonstrated good overall feasibility, suitability and acceptability. Table 11 displays evidence demonstrating the extent to which success criteria were met and potential improvements to trial design.

## Discussion

### Principal findings

This feasibility trial evaluated the feasibility, suitability and acceptability of assessing the effectiveness of independent prescribing by APPs for patients with LBP in primary care, to inform the design of a future definitive SWcRCT. Over a recruitment period of 6 months, 29 participants were recruited across three clinical sites. The average age range of participants was 40–49 years, reflective of international demographic data for LBP [92–94]. Trial objectives were evaluated against predefined success criteria. 90% of the success criteria were met. Specific objective benchmarks evaluating adequate time to complete 'trial-related tasks' and recruitment and retention targets were not met. 48% of participants were lost to follow up by 6 weeks with 65.5% lost to follow up by 12 weeks. Both the planned primary and secondary outcome measures were feasible and acceptable.

### Acceptability of interventions

90% of the success criteria were met indicating that the methods tested are feasible, suitable and acceptable for use in a definitive trial. The data further strengthens trends found in the literature, demonstrating that healthcare service users are accepting and satisfied with NMP and have confidence in clinicians' NMP skills and competence [95, 96]. Specifically, participants welcomed the APP's ability to include prescribing as one part of a comprehensive and holistic management plan. The clinicians all agreed that NMP enables better holistic care by FCPs for LBP in primary care.

### Eligibility criteria

The eligibility criteria were designed to enable the recruitment of patients experiencing medium risk LBP, stratified by the STarT back tool. The majority of patients stratified to the medium risk group are acute or subacute in nature, potentially benefitting from a multi-modal management approach including the use of analgesia [5, 6]. APPs participating in the feasibility trial agreed that the eligibility criteria were suitable to allow the evaluation of the trial objectives. They echoed the literature in recognising that the condition is the predominant musculoskeletal problem presenting at primary care clinics potentially requiring analgesia as part of its management [5, 56–58]. Synthesis of findings support amendment of the eligibility criteria for a full trial to include LBP patients across all three STarT back stratification groups.

Epidemiological literature highlights that for first episodes of LBP, pain is seen to improve rapidly in the first four to six weeks and is commonly fully resolved by 12 weeks [94, 97]. This is not the case for the majority of patients with recurrent episodes or persistent LBP. In these

**Table 10. Focus group, comments that reported or discussed each theme and illustrative quotations from patients (quotations have been copied verbatim).**

| Theme | Illustrative Quotations |
|---|---|
| The use of physiotherapist independent prescribing in FCP clinics | "When you go to a doctor and say, "I've got back pain," they'll sort of say, "Right. . . I'll give you some painkillers to take for a couple of weeks,", Whereas if you're with [APP 2}, she will say. . . "do X, Y, Z and if you can't manage it, then come back and we'll try something," and vice versa. So, you get more information on how the drugs will work for you if you need them. . ..." (Participant 3) |
| | "The general practitioner will try to refer on if they're not sure what's going on. Whereas your MSK consultant [FCP] is looking after your pain, your physiotherapy and your forward treatment. I think it's good in the one package." (Participant 2) |
| | "It seems you're getting a solution to the problem rather than having to wait and wait and wait and see other people that you have to explain the same thing to every time you meet them." (Participant 6) |
| | "Sometimes, you need medication to take the pain away so that you can do strengthening exercises and then when you go back, you've got a better range of movement, so you don't need the tablets, you know, whichever way round." (Participant 1) |
| | "If you're pain free, you can get back to work, they [APPs who can prescribe] get you back to work as quickly as possible." (Participant 5) |
| | ". . . and they're [GPs] sitting there typing, "Okay, right. Well, just take the co-codamol for a couple of weeks," you know, sort of thing. I don't mean it as harshly as that but that's how it is, it can be." (Participant 4) |
| | "It's more of a holistic view." (Participant 1) |
| | "It [LBP] is like a specialist subject. . .. . . it's not really suited to general practice" (Participant 2) |
| | "If it's soft tissue, they [GPs] send you off for an MRI or CT and say, "We'll give you a referral in three weeks' time," whereas the APP would say, "I think it's this. I want to give you treatment for this and we're going to give you some medication, some exercises and follow-on care." (Participant 2) |
| | "I love doctors, don't get me wrong but as I say, it's like a 10 minute [appointment] and they don't really know you and it's, "Oh well, I'll just write you out a prescription," . . .." (Participant 4) |
| Trial conduct and processes | "I think so long as it's explained to the patient in the beginning that it is a trial and you have to complete it. Even if you've got better in the middle of it, you've still got to fill in the surveys to say, "I got better". Because a lot of people don't bother." (Participant 2) |
| | ". . .ask the patient initially how they would like to be contacted, you know, whether they would mind having a reminder call of some description. . .. . ." (Participant 4) |
| | ". . .. some folks are social media savvy and would be okay to contact them by email or by MSN. Or, if they joined a closed group on a Facebook site, where there was a community and reminders came on that. But for an older patient, it could be difficult to interface. If they don't have a computer themselves, they'd have to rely on someone else logging on for them. . .. . ." (Participant 5) |
| | ". . .. . a lot of patients, especially the older ones just don't go on the internet, don't want to know anything about it. So, you'd have to have a different way of communicating." (Participant 3) |
| | ". . . the wider sort of rurality. You know, so you've still got have options. You do when you join a website, they say, "How would you like to be contacted?" So, if you've got all the options, you can pick one or all of them and you can always change them at any time." (Participant 1) |
| | "Well, it obviously doesn't work for everyone, but you could have a liaison point in the surgery." (Participant4) |
| | "I don't know whether it [the trial] needs a proper clinic advert/poster. I mean, I know the receptionists tell. . . and obviously, people say by word of mouth but just that extra. . ." (Participant 6) |
| | "[to aid in recruitment] . . .. What about the use of the television screen in the waiting room?" (Participant 5) |
| | "[To promote retention in the trial] . . .. I would just sell the message that completing the survey is not about an individual case, it's about back pain." (Participant 1) |
| | "I think so long as it's explained to the patient in the beginning that it is a trial and you have to complete it. Even if you've got better in the middle of it, you've still got to fill in the surveys to say, "I got better." Because a lot of people don't bother." (Participant 3) |
| | "Well, I think it's people [those not completing follow up surveys] who, if they get better, they think they don't need to carry on." (Participant 5) |

*(Continued)*

**Table 10.** (Continued)

| Theme | Illustrative Quotations |
|---|---|
| Data Collection | "You just got used to it and you forgot it was there and mine didn't roll or peel, you know, because you gave me a spare dressing in case it rolled up but it was absolutely fine; you forget it's there." (Participant 1) |
| | "I was walking like five, seven miles a day and I was swimming probably two miles, three miles a week, something like that, with it on." (Participant 6) |
| | "I have very sensitive skin so when I first put it on, I wondered if I'd get eczema, because that's what I suffer from. Not a bit of it, just forgot about it. And it came off easily in the end." (Participant 2) |
| | "Well, that was another thing that I remember flashed across my mind at the time. If it had come off, did it matter where it was put back on? You know, does it have to go back in exactly the same place, or could you move it?" (Participant 3) |
| | "It did for about the first couple of days but then you forget about it, so I thought, "Well, there's no point deliberately doing anything," because that's not giving a true thing." (Participant 1) |
| | "No, it didn't worry me. It's just that I wasn't quite sure what its purpose in life was, if you see what I mean. . . . . . I'd forgotten what had been explained to me!" (Participant 4) |
| | I'm sure that it was explained to me, otherwise I wouldn't have had it attached to my person. . . . . .. but when I got home, I thought, "What is it doing on my leg? |
| | And why is it doing it?" you know, it had just sort of gone over the top by the time I'd got home." (Participant 2) |
| | "It was the one [digital VAS] where. . . yeah, we had when it was vertical. It just kept moving up and you couldn't do it. So, that would probably be better horizontal." (Participant 6) |
| | "I don't like questions that have like a scale. I can't think what percentage out of 100 is. . . "(Participant 1) |
| | ". . .one to 100 is quite a big range. At least one to 10, you've got so many points you can think of. . ." (Participant 3) |
| | "You need a timescale of some description as to what point of the day you're answering that questionnaire. Because I know I did mine at night. Well, at night, my back pain is absolutely horrible. If I'd done it a 9:00 in the morning, you'd have probably got different answers" (Participant 4) |
| | "I was filling it in at work and I had to take a phone call and then I came back and there was a problem with it, I couldn't restart it. I had to start it from the beginning again." (Participant 6) |

patient groups, pain is often accompanied by more prominent psychosocial drivers, with patients commonly developing issues which require a clear long-term individualised psychosocial management plans [94, 97]. If a definitive trial is to include all STarT Back stratification groups, additional longitudinal follow up procedures should be incorporated, rescheduling the trial primary endpoint to 1 year to allow for evaluation of patients in the long term.

The eligibility criteria specified that only patients with non-specific back pain +/- leg pain requiring medication advice and drug prescription qualified for inclusion. Findings highlighted that the scope of iNMP includes not only the prescription of medicines but de-prescribing and medicines advice and management. Clinicians advised that these key skills are the prescribing skills most frequently optimised across the spectrum of LBP. The NHS spent £17.4 billion on prescription medications in 2016/17, with prescribing of analgesia for MSK pain significantly increased compared to the previous decade [98]. The APPs reported that they are frequently required to de-prescribe inappropriate and/or potentially harmful analgesia provided by other clinicians, or to optimise the use of drugs already prescribed to enhance rehabilitation potential. These observational findings emphasise the inappropriate overuse of paracetamol, non-steroidal anti-inflammatory drugs (NSAIDs), opioids and gabapentinoid medications for the treatment of pain reported in the literature, despite published prescribing guidelines [6, 99]. In the UK, 24 million prescriptions for opioids were issued in 2017 [100], with gabapentinoid prescribing tripling over the last decade. Many of these drugs were prescribed for persistent LBP +/- leg pain [101]. Although it is hoped that these drugs are prescribed appropriately within governance frameworks, it is postulated that repeat prescriptions alongside insufficient clinical follow up, propagate prolonged use of these potentially dangerous drugs [102]. There is a current deficit in research evaluating how physiotherapist

**Table 11. Success criteria.**

| General Objectives | A Priori Success Criteria | Achieved Yes/No | Evidence/Comments |
|---|---|---|---|
| Eligibility criteria | A favourable number of patients fit the eligibility criteria to enable the stipulated recruitment rate | Yes | The eligibility criteria were reported as suitable and acceptable by all recruiting APPs and enabled a feasible recruitment rate. Barriers to recruiting all eligible patients are reported in the qualitative component synthesis. |
| | APPs agreed with the eligibility criteria | Yes | All APPs agreed with eligibility criteria for the feasibility trial. Qualitative data highlights that all STarT back stratification groups should be included in a full trial. |
| Recruitment strategy | Participants were recruited within the time constraints of the local clinical environment | Yes | All participants were recruited within clinic time constraints. |
| | | | However, APPs felt the time pressures were stressful and recommended increasing appointment times or use of research assistants. |
| | Patients and APPs report that they were happy with the recruitment strategy | Yes | The recruitment strategy was deemed acceptable to both patients and APPs. However, APPs cited a lack of time as the major challenge to recruitment, recommending the use of research assistants in a full trial. |
| Data collection methods | Data were collected with ease via REDCap and no complications were experienced | Yes | REDCap collected the data well with no errors. |
| | Data completeness of ≥ 80% | Yes | 100% data completeness was achieved. |
| | Patients and APPs report that they were happy with the data collection methods | Yes | Patients and APPs deemed all data collection methods acceptable. It was highlighted that the REDCap application was "clunky" on the tablet, therefore investment in higher spec tablets for a full trial was recommended. Participants also recommended the use of horizontal VAS scales over vertical due to difficulties with screen scrolling. |
| Follow up procedures | 100% of participants were contacted for follow up | Yes | 100% of participants were contacted. |
| | ≥80% completion of follow up outcome measures | No | Loss to follow up:<br>• 6 weeks, 48%<br>• 12 weeks, 65.5% |
| | Patients and APPs report that they were happy with follow up procedures | Yes | Patients and APPs reported that the follow up procedures were acceptable. However, recommended reminder telephone calls and the option to complete the follow up outcome measure surveys: digitally, on paper, over the telephone, via video call or face-to-face with a member of the research team. |
| **Specific Objectives** | **A Priori Success Criteria** | **Achieved Yes/No** | **Evidence/Comments** |
| *Feasibility* | | | |
| Participant recruitment rates | Recruitment target of n = 10 per clinician met in the time available (6 months) | Yes x2 | At x2 sites the stipulated n = 10 participants were recruited within the 6month recruitment window. |
| | | No x1 | At the capital city site n = 9 participants were recruited. Reasons for slower recruitment are described in the synthesis of the interview data. |
| Ease of fitting accelerometers | Accelerometers were fitted within the allocated clinical time allowed with the FCP APP | No | Additional time or use of research assistants was recommended by the APPs. |
| | Patients and APPs report that accelerometers were fitted with no issues | Yes | No issues or adverse reactions were reported. |
| Accelerometer data collection | REDCap was able to capture the data from the accelerometers with no errors or data loss | No | Specific ActivPal applications were required to collect and store accelerometer raw data. Once downloaded, the data was transferred to the university server, as per ethical approval. |
| | Patients report that they were happy with data collection using accelerometers/ burden within subjectively appropriate limits | Yes | Patients all reported that they were happy with the use of accelerometery and felt no increased burden. |
| Capacity (time and effort) of clinicians' complete trial related tasks | APPs report that adequate time was allowed to complete all tasks required by them during the trial | No | APPs reported significant time pressures, recommending the use of research assistants or increased appointment times in a full trial. |

*(Continued)*

**Table 11.** (Continued)

| | | | |
|---|---|---|---|
| Training requirements required by clinicians | APPs report that they had adequate training to be able to complete the tasks required by them during the trial | Yes | APPs all felt adequately trained, however identified that less experienced clinicians would require training around ethical consent. All APPs in a full trial would require training for fitting and using accelerometers. |
| *Suitability* | | | |
| Outcome measures | Data completeness of ≥ 80% | Yes | 100% achieved. |
| | Patients and APPs report that the outcome measures were appropriate and self-explanatory | Yes | Patients and APPs stated that the outcome measures were suitable. APPs advised the use of a sleep and physical activity questionnaire to accompany accelerometer data. Patients stated that some participants in a full trial might require help interpreting questions dependent on literacy levels. |
| Compliance with wearing the accelerometers | Data collected ≥ 80% of the requested time (16hrs/day for 7 days) | Yes | 100% compliance achieved. |
| Time required to conduct each stage of the protocol | APPs report having adequate time to complete each stage of the protocol | No | More time required to recruit, consent and fit accelerometers recommended. |
| Service infrastructure | Recruitment targets met. | No | One site did not attain the recruitment target. |
| | Data completeness of ≥ 80% | | 100% data completeness was achieved. |
| | APPs report that adequate service infrastructure is in place to allow for a full trial to be completed | Yes | Infrastructure was described as suitable. |
| *Acceptability* | | | |
| Intervention | Patients and APPs report that the intervention was appropriate/ satisfactory | Yes | All participants in the qualitative component reported that the intervention was acceptable, appropriate and satisfactory. |

independent prescribing is used to manage LBP. It is imperative that a full trial collects and evaluates data inclusive of the whole scope of physiotherapist independent prescribing (including what is prescribed, de-prescribed or advised), with eligibility criteria enabling the inclusion of all patients with non-specific back pain +/- leg pain.

## Recruitment

Recruitment rates were found to vary between the sites where identical weekly appointment slots were available. The rural and regional city sites took approximately one month to recruit 10 participants with the capital city site recruiting nine participants over the full six-month recruitment period. Interestingly, the key reason identified for the slower recruitment rate was that patients fulfilling the eligibility criteria at this location did not want to take medication for their pain. Instead, participants reported that access to a physiotherapist for assessment and management permitted an alternative to the pharmacotherapy provided by the GP. This reflects the literature evaluating the use of direct access to physiotherapy in primary care. Physiotherapeutic holistic assessment for MSK conditions and joint decision-making regarding the appropriate management plan for the individual, has been shown to provide greater levels of patient satisfaction when patients are able to seek care directly from a physiotherapist without prior mandatory medical-input [103, 104]. Completion of an adequately powered trial would be feasible using recruitment rates based on the rural and regional sites not that obtained in the capital city. However, it is posited that with the expansion of the proposed eligibility criteria to include all patients with non-specific LBP +/- leg pain, the full scope of physiotherapist prescribing and the adoption of additional recruitment capacity via research assistants and administrative staff, that recruitment rates and retention at all sites would be acceptable.

## Follow up procedures and retention

Poor clinician time capacity is a recognised barrier to conducting clinical trials [44, 105, 106]. Both patients and APPs advocated for the use of research assistants to aid with trial

recruitment, consent and follow up administration. To improve retention, participants recommended reminder telephone calls and one-to-one appointments where participant literacy levels limited completion of follow up surveys. Further, it was proposed that research assistants would improve retention by acting as a consistent point of communication, encouraging smooth participant flow through the trial.

Previous literature has linked the use of a combination of recruitment and follow up strategies, with improved retention rates [105, 107]. This improvement is attributed to sustained, frequent contact with participants as they move through a trial. Adequate statistical power and good external validity rely on sufficient participant numbers [44, 105, 106]. As loss to follow up in this feasibility trial was higher than the 20% deemed acceptable within research methods literature [105, 107], it is essential that the design of a full trial engages several strategies to improve retention. The literature proposes that a minimum of three communication channels should be provided by each participant, including contact through friends and relatives, with regular updates, 'check in' communication via text, email and telephone and face-to-face appointments if preferred. 'Branding' the trial with a recognisable name and logo embossed on all trial documents and correspondence may also improve retention owing to inferred credibility, enabling participants to build a bond with the research [105, 107].

## Outcome measures and data collection methods

The literature reports that the use of a core outcome set assessing pain intensity, health related quality of life and physical function is required for the assessment of non-specific LBP [28]. As optimal tools are not defined in the literature, the primary and secondary outcome measures were selected and agreed upon by a group of clinical and academic experts. The appropriateness of the selected outcome measures for use in a full trial were evaluated. All APPs and patients agreed that the outcome measures were suitable and acceptable. However, the patients recommended that the option of completing the survey on paper would be beneficial for those with limited access to email or poor IT skills. Data demonstrated graduated improvements in pain, function, disability and activity over the 12-week assessment period, mirroring the trends for medium and low risk LBP reported in the literature [92, 93, 97]. No floor effects were detected across the outcome measures used. 100% data completeness was achieved by using an online survey. Data collection via an online survey was deemed acceptable and feasible by both the clinicians and patients, supporting the literature that demonstrates better and quicker response times with fewer missing responses across both open and closed survey questions [108, 109].

This feasibility trial aimed to evaluate participant compliance through assessments of wearing an accelerometer alongside the ease of fitting the devices and data collection. Treatment effect was not assessed. Participants fitted with accelerometers achieved 100% compliance and 100% data completeness, demonstrating feasibility of use in a full trial. All participant returned the accelerometer in person, therefore the feasibility of returning the device by post was not formally evaluated. Participants and APPs reported that the devices would be useful to include in a full trial. They were fitted easily and owing to their positioning participants did not feel that the devices prompted them to increase activity levels after the first 24hrs. This is consistent with the accelerometer literature which demonstrates that removable accelerometers worn on the wrist, ankle or clothing may prompt increased physical activity and might lead to poor data completeness owing to participant removing the devices and forgetting to replace them [110–112].

The published protocol for the feasibility trial detailed the assessment of sleep via accelerometer data. Unfortunately, this was not evaluated owing to restriction in the technology available for testing [113]. This deficit was highlighted by the APPs, who recommended the addition of a questionnaire-based outcome measure assessing participants' sleep. 50–60% of people experiencing either acute or persistent low back pain experience high levels of sleep disturbance [114]. Poor sleep over long periods of time may lead to depression, obesity, diabetes and cardiovascular disease [114, 115]. Patients with LBP suffering with sleep disturbance have been reported as twice as likely to be hospitalised owing to their pain [116]. The literature demonstrates that improved sleep modulates pain intensity [117], with poor quality sleep associated with increased pain intensity, fatigue, decreased function and psychological stress. Although perceived sleep quality has been shown to be different to the objective reality assessed via polysomnography or actigraphy, subjective assessment via sleep questionnaires and diaries have been shown to be valuable where objective evaluation is not possible [118]. Based on this rationale, it would be suitable to add a validated and reliable sleep questionnaire into the outcome measures survey for use in a full trial.

Findings from this feasibility trial indicate that a definitive SWcRCT is feasible following some minor modifications. The SWcRCT should include all patients presenting with non-specific LBP +/- leg pain and capture data representative of the full scope of physiotherapist independent prescribing. To navigate limited clinician capacity and time restrictions dictated by job plans and service specifications, researchers should consider the use of research assistants to recruit, consent, aid in data collection and complete follow-up and administrative tasks. Prior to the completion of a definitive full SWcRCT, recruitment and follow up procedures should be modified in accordance with the feasibility trial data. The online outcome measures survey should be revised to include a validated sleep evaluation tool, and the survey logic updated. Revised procedures and both online and paper versions of the survey should be piloted across all LBP stratification groups to evaluate successful modification before use in a definitive full SWcRCT.

## Strengths and limitations

This feasibility trial used rigorous systematic methods including analysis and synthesis strengthened by an imbedded qualitative component and the engagement of expert trial management and steering groups including clinicians, healthcare managers, academics and patient and public representation. This combination ensured specialist knowledge of physiotherapist independent prescribing and LBP alongside specific primary care perspectives, facilitating a rigorous analytical process. There were no adverse effects to the treatments or methods evaluated. Individuals recruited to the qualitative component of the trial were observed to be comfortable throughout the process, expressing their thoughts and opinions openly. This feasibility trial is limited by the small samples used in both the trial and qualitative components; however, samples did satisfy the theoretical representation of the population essential to evaluate the trial objectives. No guidelines exist defining best practice for feasibility trials evaluating trial methods prior to SWcRCT. Although this may limit the trial, the authors have utilised transparent, integrated best practice from aligned guidelines, whilst ensuring robust consultation with subject and methodological experts and representatives from the public, throughout trial design.

## Conclusion

A definitive SWcRCT is feasible with some minor modifications. Methods evaluated are feasible, suitable and acceptable for use in a definitive SWcRCT. The SWcRCT should include all

patients presenting with non-specific LBP +/- leg pain and capture data representative of the full scope of physiotherapist independent prescribing. Research assistants should be used to overcome limited clinician capacity.

## Supporting information

**S1 File. Published protocol.**
(PDF)

**S2 File. Outcome measures questionnaire.**
(DOCX)

**S3 File. Participant information sheet.**
(DOCX)

**S4 File. Participant consent form.**
(DOCX)

**S5 File. Topic guide for interviews.**
(DOCX)

**S6 File. Topic guide for focus group.**
(DOCX)

**S7 File. Success criteria.**
(DOCX)

**S1 Checklist.**
(DOC)

**S2 Checklist. CONSORT checklist.**
(DOC)

## Author Contributions

**Conceptualization:** Tim Noblet, John Marriott, Alison Rushton.

**Data curation:** Tim Noblet, Alison Rushton.

**Formal analysis:** Tim Noblet, Alison Rushton.

**Funding acquisition:** Tim Noblet, Alison Rushton.

**Investigation:** Tim Noblet, Alison Rushton.

**Methodology:** Tim Noblet, John Marriott, Alison Rushton.

**Project administration:** Tim Noblet.

**Resources:** Tim Noblet, John Marriott, Amanda Hensman-Crook, Simon O'Shea, Sarah Friel, Alison Rushton.

**Supervision:** Alison Rushton.

**Validation:** Alison Rushton.

**Writing – original draft:** Tim Noblet.

**Writing – review & editing:** John Marriott, Alison Rushton.

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
