## [Decision Letter · Decision Letter 0]

4 Dec 2019

PONE-D-19-31324

Independent Prescribing by Advanced Physiotherapists for Patients with Low Back Pain in Primary Care: a feasibility trial with an embedded qualitative component

PLOS ONE

Dear Mr Noblet,

Thank you for submitting your manuscript to PLOS ONE. After careful consideration, we feel that it has merit but does not fully meet PLOS ONE’s publication criteria as it currently stands. Therefore, we invite you to submit a revised version of the manuscript that addresses the points raised during the review process.

Please be aware that responding to the revisision does not guarantee eventual acceptance of your manuscript. In your revision you should make every effort to address the concerns raised by the reviewers. If you disagree with a suggestion or change recommended by the reviewer, you should make clear the rationale behind your position and include this in the letter to the editor which accompanies the revision.

We would appreciate receiving your revised manuscript by Jan 18 2020 11:59PM. To enhance the reproducibility of your results, we recommend that if applicable you deposit your laboratory protocols in protocols.io, where a protocol can be assigned its own identifier (DOI) such that it can be cited independently in the future. For instructions see: http://journals.plos.org/plosone/s/submission-guidelines#loc-laboratory-protocols

We look forward to receiving your revised manuscript.

Kind regards,

Johannes Fleckenstein

Academic Editor

PLOS ONE

Journal Requirements:

2. Thank you for submitting your clinical trial to PLOS ONE and for providing the name of the registry and the registration number. The information in the registry entry suggests that your trial was registered after patient recruitment began. PLOS ONE strongly encourages authors to register all trials before recruiting the first participant in a study.

i) your reasons for your delay in registering this study (after enrolment of participants started);

ii) confirmation that all related trials are registered by stating: “The authors confirm that all ongoing and related trials for this drug/intervention are registered”.

Please also ensure you report the date at which the ethics committee approved the study as well as the complete date range for patient recruitment and follow-up in the Methods section of your manuscript.

3. Your ethics statement must appear in the Methods section of your manuscript. If your ethics statement is written in any section besides the Methods, please move it to the Methods section and delete it from any other section. Please also ensure that your ethics statement is included in your manuscript, as the ethics section of your online submission will not be published alongside your manuscript.

Additional Editor Comments (if provided):

Reviewers' comments:

Reviewer's Responses to Questions

**Comments to the Author**

1. Is the manuscript technically sound, and do the data support the conclusions?

Reviewer #1: Yes

Reviewer #2: Yes

Reviewer #3: Partly

2. Has the statistical analysis been performed appropriately and rigorously? 

Reviewer #1: N/A

Reviewer #2: Yes

Reviewer #3: N/A

3. Have the authors made all data underlying the findings in their manuscript fully available?

Reviewer #1: Yes

Reviewer #2: Yes

Reviewer #3: Yes

4. Is the manuscript presented in an intelligible fashion and written in standard English?

Reviewer #1: Yes

Reviewer #2: Yes

Reviewer #3: Yes

5. Review Comments to the Author

Reviewer #1: The authors present a comprehensive report of a feasibility trial of Independent prescribing by advanced physiotherapists for patients with low back pain, which addresses an increasingly recognised clinical need.

There are a number of sections where additional detail would strengthen the manuscript and its overall conclusions as outlined below.

Introduction

In the Introduction the authors have presented a good rationale for the need for the trial from the current evidence base but have not given any justification for the stepped wedge trial design proposed which first appears in the Methods. A brief rationale for this trial design in the Introduction would strengthen this section.

Similarly, within Table 1 specific objectives under Feasibility it is proposed to evaluate the ease of fitting participants with accelerometers but there is no explanation within the Introduction section to justify this. A brief review of the literature on this topic would again strengthen the Introduction.

There are also no objectives related to adverse events associated with prescribing and I would be interested to know if this was considered by the authors.

Methods

Outcome measures - please clarify how participants unable to complete these measures online or due to literacy issues were dealt with and the implications of excluding people for these reasons alone

Accelerometer - Provide more details of how 10 participants were selected and the Accelerometer type, data being collected and how it was fitted. Justify why there was only one data collection point as presumably in the definitive trial they will be used at follow-up and you have not tested the feasibility of this data collection method.

Participants - state how and who identified interested patients and the stage in their management they were identified.

Intervention - clarify if participants had been prescribed medications by their GP prior to entry into the study. More detail of the protocol for the intervention is needed as it is not clear how it was operationalised or documented in practice regarding medication advice/prescribing/repeat prescribing. Which medications did the APP have a license to prescribe? What additional training or insurance was required to include this within their clinical practice?

Outcomes - clarify the restriction in technology that prevented sleep measurement and why another measure [ie sleep diary] was not used. Table 3 presents a detailed rationale for a wide range of measures - it would be useful to include some of this detail in the text to justify their inclusion, i.e. Tampa Scale of Kinesiophobia

Sample size -the target recruitment rate of 10 participants per APP over a 6 month period seems very low and unrealistic for testing feasibility for a definitive trial. Please provide more justification for this.

Qualitative Studies

APP interviews - could the authors provide the interview schedule for these interviews for consistency with the Focus Group topic guide which is provided

Integration - within the methods reference is made to the a priori success criteria -but these are not evident until further into the Results section - reference to Table 10 and/or a short synopsis in the text would be useful at this point in the manuscript.

Results

The recruitment rate of two practices appears notably higher than the target 6 months - could you clarify this in the text

Flow chart - please check the n= under Fulfilled eligibility criteria - as it states 7 participants were excluded but 29 were allocated to the Intervention - should this be 36?

Table 4 - add details of the employment status of participants

In reporting loss to follow-up rates please state the expected rates were 80% a priori in the text

Table 5 - clarify the variables being reported under EQ5D 5L - Health today appears to be using a 0-100 VAS score which is declining over time indicating a worsening health state?

Clarify the number of participants the work absence data relates to

Accelerometry

Add the age/gender of n=10 participants and the sites they were recruited from to Table 6.

Clarify if there were any missing data or non-wear times

Qualitative Interviews - APPs

Please provide some demographic details of the three APPs - age, level of experience, current job title and postgraduate qualifications

Follow-up procedures- the need for research assistants to recruit in a definitive trial is evident from these interviews. Could the authors address the issue of blinding of outcome assessors and how this would be addressed in a definitive trial if face-to-face outcome assessment is used.

Recruitment - could the authors discuss how selection bias was minimised in this feasibility trial as APPs were recruiting

Follow-up - As all participants returned Accelerometers in person the trial has not demonstrated the feasibility of using postage as a follow-up method which should be acknowledged in the text.

Qualitative Interviews - Patients

Clarify if the n=6 participants were compliers or drop-outs and if compliers, how reasons for drop-out will be identified and dealt with in the definitive trial. Clarify also if all three clinics were represented in this group of participants.

Discussion

Eligibility criteria -

Discuss the implications of including all three STarT back subgroups in a definitive trial as proposed. What is the proposed sample size for a definitive trial?

State the proposed long-term follow-up time point(s) being considered for a definitive trial.

Explain how the prescribing practice of APPs was recorded in this feasibility trial and would be recorded in a definitive trial as this is not evident from the data presented and is a key variable to report to demonstrate APPs prescribing practice.

It would also be informative to include literature on the views of GPs on APPs taking on a prescribing role and how it is proposed this role will be shared in future practice.

The statement that a definitive trial is feasible with minor modifications needs some tempering as there are alot of proposed changes to the trial protocol including increasing its eligibility criteria, increasing the number of outcome measures, follow-up methods and follow-up points and providing additional training of APPs and recruiting and training research assistants to name a few, some of which will need to be piloted. I would consider these to be more than minor modifications and would encourage the authors to revise this wording in the Discussion, Conclusions and Abstract.

Reviewer #2: To my opinion, this is an excellent manuscript about an interesting study. However, I recommend the authors to look closer on following: 1) ensure that all abbreviations are explained the first time they occur; 2) help the readers to interpret the primary and secondary outcomes measure data, both in the text and in the Table 5; 3) demographic data of the participating APP seem to be missing.

Reviewer #3: Thank you for inviting me to review this study.

I have a number of major concerns about this study:

1. The authors seem to have tried to combine a desire to pilot their future trial and assess the feasibility of the intervention... however, their design only permits them to assess the feasibility of their intervention... I can understand that the stepped wedge component may have been difficult to do but I was unclear why a cluster feasibility pilot trial design was not undertaken? If the objective was not to pilot the future trial but to only examine the feasibility of intervention - then its unclear why the authors report success criteria for the recruitment and retention rates?

2. The authors have chosen an RCT design to test an intervention of something that is not a new or novel intervention, but actually a different professional role thats part of a new service delivery approach - thats very specific to the NHS in the UK. I was therefore uncertain about the wisdom of a trial design and of the international relevance of the study. I wonder if a service evaluation might have been a better design?

3. If the authors feel this is a new intervention worth testing for effectiveness, then some background information on how it was developed and the key targets of the intervention, what might constitute a per protocol intervention and what the training and competencies are for the intervention - would all be helpful. At present there are 2 sentences on it.

4. It was unclear how and where patients were recruited and identified and from what denominator of those invited?

5. A major concern for cluster trials is recruitment selection bias. This was not mentioned.

6. What was the purpose of collecting accelerometer data in relation to the feasibility study? Was that a key treatment target? Or was that actually piloting an outcome for the main trial?

7. The authors collected a range of outcome measures they plan to use within a future main trial – but these seemed to be primarily to be about piloting them for the main trial and were largely unrelated to questions about the feasibility of the intervention. For example, there was no metrics to determine fidelity to the intervention, safety, adverse reactions or perceived harms, or patient credibility and confidence of the intervention…

8. The qualitative aspects seem to lack any theoretical framework…

9. In the results it appears that physical function decreased over 3 months and that anxiety and depression worsened…. How do these disappointing results suggest feasibility or any promise from the intervention?

10. The attrition rates at 12 weeks were awful…. unacceptably low.

11. Table 5 n= values are missing and I wondered why the EQ-5D utility score was not given?

12. For an intervention study about prescribing - the measure called "Total prescription utilisation" seems a very blunt and inadequate measure… Could we not know what APPs prescribed for these patients? Perhaps provided in categories such as NSAIDs, opioids, atypical analgesics etc…

Overall, I found the paper very long and confusing with lots of Tables.

6. PLOS authors have the option to publish the peer review history of their article (what does this mean?). If published, this will include your full peer review and any attached files.

Reviewer #1: Yes: Deirdre Hurley

Reviewer #2: No

Reviewer #3: No

---

## [Author Response · Author response to Decision Letter 0]

23 Dec 2019

Please see attached document for detailed response to reviewers. Thank you again for your time.

---

## [Decision Letter · Decision Letter 1]

30 Jan 2020

PONE-D-19-31324R1

Independent Prescribing by Advanced Physiotherapists for Patients with Low Back Pain in Primary Care: a feasibility trial with an embedded qualitative component

PLOS ONE

Dear Mr Noblet,

Thank you for submitting your manuscript to PLOS ONE. After careful consideration, we feel that it has merit but does not fully meet PLOS ONE’s publication criteria as it currently stands. Therefore, we invite you to submit a revised version of the manuscript that addresses the points raised during the review process.

We would appreciate receiving your revised manuscript by Mar 15 2020 11:59PM. To enhance the reproducibility of your results, we recommend that if applicable you deposit your laboratory protocols in protocols.io, where a protocol can be assigned its own identifier (DOI) such that it can be cited independently in the future. For instructions see: http://journals.plos.org/plosone/s/submission-guidelines#loc-laboratory-protocols

We look forward to receiving your revised manuscript.

Kind regards,

Johannes Fleckenstein

Academic Editor

PLOS ONE

Reviewers' comments:

Reviewer's Responses to Questions

**Comments to the Author**

1. If the authors have adequately addressed your comments raised in a previous round of review and you feel that this manuscript is now acceptable for publication, you may indicate that here to bypass the “Comments to the Author” section, enter your conflict of interest statement in the “Confidential to Editor” section, and submit your "Accept" recommendation.

Reviewer #1: (No Response)

Reviewer #3: (No Response)

2. Is the manuscript technically sound, and do the data support the conclusions?

Reviewer #1: Yes

Reviewer #3: (No Response)

3. Has the statistical analysis been performed appropriately and rigorously? 

Reviewer #1: Yes

Reviewer #3: (No Response)

4. Have the authors made all data underlying the findings in their manuscript fully available?

Reviewer #1: Yes

Reviewer #3: (No Response)

5. Is the manuscript presented in an intelligible fashion and written in standard English?

Reviewer #1: Yes

Reviewer #3: (No Response)

6. Review Comments to the Author

Reviewer #1: The authors have addressed the majority of my comments comprehensively. There are a few minor changes related to three responses that would increase the accuracy of the manuscript as detailed below:

Point 2 - the last sentence in the response is not an accurate reflection of the literature as the feasibility of using ActivPal accelerometers with LBP patients has been previously tested in the following trial.

McDonough SM, Tully MA, Boyd A, O'Connor SR, Kerr DP, O'Neill SM, Delitto A,

Bradbury I, Tudor-Locke C, Baxter GD, Hurley DA. Pedometer-driven walking for

chronic low back pain: a feasibility randomized controlled trial. Clin J Pain.

2013 Nov;29(11):972-81. doi: 10.1097/AJP.0b013e31827f9d81.

Point 8 - The response concerning the limitations in validated accelerometer technology does not explain the issue that prevented the recording of sleep data. Further rewording is recommended.

Point 15. The reporting of mean recorded data for sitting, standing and walking equates to 24hrs and does not take account of sleep time. Linked to the previous response there is a need to state how sleep can be differentiated from awake time for other researchers using this device.

Reviewer #3: (No Response)

7. PLOS authors have the option to publish the peer review history of their article (what does this mean?). If published, this will include your full peer review and any attached files.

Reviewer #1: Yes: Deirdre Hurley

Reviewer #3: No

---

## [Author Response · Author response to Decision Letter 1]

9 Feb 2020

See attached response to reviewers doc.

---

## [Editor Report · Decision Letter 2]

14 Feb 2020

Independent Prescribing by Advanced Physiotherapists for Patients with Low Back Pain in Primary Care: a feasibility trial with an embedded qualitative component

PONE-D-19-31324R2

Dear Dr. Noblet,

We are pleased to inform you that your manuscript has been judged scientifically suitable for publication and will be formally accepted for publication once it complies with all outstanding technical requirements.

With kind regards,

Johannes Fleckenstein

Academic Editor

PLOS ONE
---

## [Editor Report · Acceptance letter]

2 Mar 2020

PONE-D-19-31324R2 

Independent Prescribing by Advanced Physiotherapists for Patients with Low Back Pain in Primary Care: a feasibility trial with an embedded qualitative component 

Dear Dr. Noblet:

I am pleased to inform you that your manuscript has been deemed suitable for publication in PLOS ONE. Congratulations! Your manuscript is now with our production department. 

With kind regards,

on behalf of

Dr. Johannes Fleckenstein 

Academic Editor

PLOS ONE